# Small Ruminant Lentivirus Infection in Sheep and Goats in North Portugal: Seroprevalence and Risk Factors

**DOI:** 10.3390/pathogens12060829

**Published:** 2023-06-14

**Authors:** João Jacob-Ferreira, Ana Cláudia Coelho, Ana Grau Vila, Delia Lacasta, Hélder Quintas

**Affiliations:** 1Animal and Veterinary Research Centre, University of Trás-os-Montes and Alto Douro (UTAD), 5000-801 Vila Real, Portugal; joao.ferreira.vet@gmail.com; 2Mountain Research Center (CIMO), Polytechnic Institute of Bragança (IPB), Campus de Santa Apolónia, 5300-253 Bragança, Portugal; helder5tas@ipb.pt; 3Servicio de Sanidad Animal, Dirección General de Producción Agropecuaria e Infraestructuras Agrarias, Consejería de Agricultura y Ganadería, Junta de Castilla y León, 47014 Valladolid, Spain; ana.grau@jcyl.es; 4Animal Pathology Department, Instituto Agroalimentario de Aragón-IA2, Universidad de Zaragoza-CITA, Veterinary Faculty of Zaragoza C/Miguel Servet 177, 50013 Zaragoza, Spain; delialacasta@gmail.com

**Keywords:** sheep, goat, lentiviruses, SRLV, seroprevalence, risk factors

## Abstract

Small ruminant lentiviruses (SRLVs) are transmitted among ovine and caprine species. This disease is a severe problem for small ruminant production, not only for animals’ well-being but also for flocks’ efficiency. The main aim of this research was to quantify the seroprevalence and associated risk factors for SRLV infection in the northern region of Portugal. Samples were collected from a total of 150 flocks, of which 129 (86.0%; 95% CI: 80.67%–91.33%) had at least one seropositive animal. Out of 2607 individual blood samples, 1074 (41.2%) were positive for SRLVs. Risk factors associated with SRLV infection were species (caprine), age (>2 years old), flock size (>100 animals), production system (intensive), food production system (milk), type of activity (professional), participation in livestock competitions (yes), replacement young ewe bought (yes), and natural feeding management (yes). This knowledge empowers the implementation of effective preventive measures. Overall, biosecurity measures should be promoted and implemented with the main aim of reducing viral transmission and reducing the prevalence of this disease. We recognise that government authorities should promote and audit voluntary control and eradication programs in small ruminant flocks in the region studied.

## 1. Introduction

Small Ruminant Lentiviruses (SRLVs) are a disease that affects ovine and caprine species caused by a group of phylogenetically co-related viruses (family Retroviridae, genus *Lentivirus*). Originally, ovine disease was associated with Maedi/Visna virus (MVV), also called ovine progressive pneumonia virus (OPPV), and caprine disease was associated with caprine arthritis-encephalitis virus (CAEV). Nowadays, SRLV infection is accepted worldwide to describe different clinical and histopathological manifestations developed by the same viral aetiology [1]. Phylogenetic studies prove that SRLVs can be divided into five genotypes, A to E, with subgroups in some [2].

Seroprevalence studies have shown that SRLV infection is present worldwide [3]. Having a heterogeneous distribution, it has significant variations between continents and even in different regions in the same continent [4]. The seroprevalence studies reported different results. These studies are difficult to compare because of differences in the sensitivity and specificity of the diagnostic tests used, as well as the criteria used to define disease and sampling [5]. Remarkably, the high prevalence of SRLVs in flocks and at individual levels in various European countries is notorious; this might be explained by the high density of the ovine population and by intensive production systems [4]. Additionally, in caprine populations, studies have shown high prevalence percentages of this infection [6].

SRLV transmission from infected progenitors to offspring may occur through milking with colostrum and milk [7,8]. This kind of transmission, though important, seems to have a minor role in spreading these viruses because offspring may be infected earlier in contact with other infected animals and not truly through milking [9,10]. Adult animals can be infected by inhaling viral particles from the secretions of infected animals [11], which might be the main possible transmission route in intensive production systems [12]. During pasturage, transmission seems to be extremely low, a fact that favours extensive production systems [13,14]. Additionally, semen seems to be a possible route of virus transmission by mating and artificial insemination techniques [15,16]. However, it is uncertain if this results in female or offspring infection [17].

SRLV infection develops as a progressive, inflammatory, and wasting disease that provokes chronic lesions that affect animals’ health and prime to austere economic losses [8]. Affected individuals are persistently infected [1]. This disease may affect different organs such as the lungs, central nervous system, mammary glands, and joints [18]. When the lungs are affected, it is common to observe tachypnoea and respiratory distress due to developed interstitial pneumonia [19]. Clinical signs are initially detected with exercise, with affected individuals remaining behind when the flock moves. Both respiratory and neurological syndromes can lead the animal to progressive cachexia and subsequent death after a long period of illness [20]. Joint disease can cause lameness because of affected carpal and tarsal joints [19]. The affection of the mammary glands results from the development of indurative mastitis [21]. Thus, animals with this syndrome are prematurely slaughtered due to suboptimal production [18]. The nervous form is less frequent and may present weakness and ataxia of the posterior limbs [22]. Clinical examination and post-mortem findings can be helpful for the veterinarian when suspecting the presence of this infection in a flock. However, an early diagnosis should not be based on these, as most affected animals are asymptomatic and may develop clinical signs late after primoinfection [23,24]. This fact makes it difficult to establish an early suspicion of the entry of the infection in a flock, delaying the diagnosis of SRLV infection that should have been established previously with laboratory tests.

Among the different laboratory methods that can be used, serological techniques, such as agar gel immunodiffusion tests (AGID) and enzyme-linked immunosorbent tests (ELISA), and molecular techniques, such as PCR and RT-PCR, can be used [25]. Blood serum is the sample of choice to perform serological tests. However, other biological samples, such as milk, can also be used [26,27]. ELISA test is a method that offers optimal results, being economical and easy to execute. Compared to ELISA, AGID tests are very specific but are less sensitive [28]. However, because of the heterogeneity of this group of viruses, the late seroconversion and the fluctuating antibody response determine important difficulties in the detection of SRLVs [29]. Molecular tests are also useful in the diagnosis, especially for early detection of infection (before seroconversion) and as a complement to previous tests [7,30]. However, low viral load in animals with latent infection and high viral genetic heterogeneity decreases PCR sensitivity. Therefore, no gold standard test for diagnosis has yet been defined [25]. In this sense, and to improve the detection of infection, a combination of different laboratory tests should be used to detect the maximum number of infected animals [24,31]. For example, some control programs resort to performing sequential tests, usually ELISA tests, followed by a confirmatory test, for example, AGID [28].

Studies that address economic losses resulting from SRLV infection are scarce, with limited and incomplete information. However, authors generally agree that these are particularly significant for small ruminant producers [32]. The harmful impact on production indices and, above all, the high rate of early culling of animals due to the development of lesions and reduced production are identified as the points of most significant economic loss for flocks [33,34]. The diversity of small ruminant flocks can influence the negative economic impacts that the disease can have. Dairy flocks seem to be the most affected by these negative impacts. In these, the development of infection can decrease the amount of milk produced by infected animals [35,36] and negatively affect quality parameters and cheese yield [37,38]. Consequently, if there is lower milk production and quality of itself as well, the offspring will also have a lower growth rate [39]. Furthermore, the health and well-being of animals affected by this disease are seriously compromised. There is, however, no major direct relationship between infection and the natural death of animals [32].

Livestock farming, specifically for sheep and goats, provides an important economic, social, and cultural contribution to human beings since the housetraining of these species. Additionally, this activity has similar importance in the northern region of Portugal. Despite the recent appearance of more industrialised farms, most small ruminant farms still carry out traditional management practices. Human activities have likely influenced the ecology of diseases such as SRLV infection [40]. It is essential to note how certain anthropogenic factors, such as international trade and husbandry practices, may play an important role in the spread of this disease. This knowledge, particularly of risk factors, can support the development of more effective control programs [41]. In many countries, veterinary health authorities have implemented eradication programs, some of them voluntary. They are generally based on (i) the removal of newborns immediately after birth; (ii) the slaughter of positive animals in periodic screenings; and (iii) the segregation of the flock into positive and negative animals [42]. So far, these have allowed an extreme decrease in the prevalence of this infection [41,43]. It is crucial to convey to livestock producers the most valuable aspects of these programs, namely, to emphasise the increase in the overall profitability of the farm [44]. In the absence of an effective vaccine or treatment, the only possible approach is to implement programs of this nature that should be encouraged worldwide. Additionally, at the livestock holdings level, they should be encouraged to prepare and implement them considering the particularities of each flock and production system.

This way, the main objective of this study was to study the seroprevalence and potential risk factors associated with SRLV infection in sheep and goat farms in the north of Portugal.

## 2. Materials and Methods

### 2.1. Data Collection

Sample size was calculated based on the list of Bragança district small ruminant flocks registered at the official animal health database, PISA.net. Sample size was calculated from the population data in 2019. Only flocks with a minimum of 20 animals per flock were included in the study. The number of animals to be sampled was estimated using the formula $n={\left(1.96\right)}^{2}p(1-p)/d^{2}$ [45]. This sample size provides a 95% confidence level for an expected prevalence of 15%. Flocks sampled were proportionally allocated according to the number of flocks in the 12 counties under study. Number of samples taken per flock was 14–19. This sample size provides a 95% confidence level for an expected prevalence of 1% per flock and allows a compromise between cost and precision of the estimates. Samples in flocks were randomly collected with aleatory numbers taken from a list of animals in each flock. Blood samples from sheep and goats aged at least six months old were collected during technical visits from official veterinarians of the local health units. Sampling procedures and laboratory tests were performed from September 2019 to February 2023. A flock was defined as SRLV-seropositive if at least one seropositive animal was present. Risk factors and health management protocols were recorded in a questionnaire in all small ruminant flocks. After reviewing the literature and identifying possible risk factors for SRLV infection, a questionnaire was made [1,46,47]. The questionnaire included 40 questions that were close-ended with choices available.

### 2.2. Serological Analysis

Blood samples (10 mL) were collected from each animal by jugular venipuncture into 10 mL tubes (Vacutainer^®^, Becton Dickinson, Plymouth, UK) with a clot activator. Blood samples were allowed to clot at ambient temperature. Then, serum was obtained by centrifugation at 200× g for 10 min and stored at −20 °C until analysis.

Serological analysis was performed at Zamora Provincial Animal Health Laboratory. Infection by SRLVs of each sample was determined by a commercial indirect ELISA test (ID Screen^®^ MVV/CAEV Indirect, Innovative Diagnostics, Grabels, France) following the manufacturer’s instructions. ELISA test is based on the use of a mixture of peptide antigens resulting in superior test performance, separating positive and negative results with high sensitivity, and detecting all genotypes (including A, B, and E) with high specificity. According to the manufacturer’s data, for a confidence interval (CI) of 95%, this test has a diagnostic sensitivity and specificity of approximately 91.7% and 98.9%, respectively [48].

### 2.3. Statistical Analysis

Data collected were recorded in Microsoft Office Excel^®^ (version 2305 Build 16. 0. 16501. 20074). Answers to the questionnaire of each farm were matched to the laboratory results through their official herd code identification, respecting the typology of the question. Variable analysis was performed using the chi-square test ($X^{2}$) to verify the association between variables. JMP^®^ Statistical Discovery version 7 software was used for this analysis. A significant effect was considered when *p* < 0.05, a very significant effect when *p* < 0.01, and a highly significant effect when *p* < 0.001. A univariate analysis was performed between the independent variables according to the association between the causes of failure and the potential risk factors. Odds ratio (OR) values were estimated, and 95% CI was calculated.

## 3. Results

### 3.1. Seroprevalence of SRLVs

A total of 150 small ruminant flocks from the north region of Portugal participated in this study. Table 1 represents the SRLV seroprevalence results from individuals and flocks as well. Overall, a serological investigation was made in 2607 samples of ovine and caprine species from a total of 150 flocks.

One hundred and twenty-nine (129) flocks had at least one animal positive for SRLVs, with an apparent prevalence of 86.0% (95%CI: 80.67–91.33%). Considering the sensitivity (91.70%) and specificity (98.90%) of the diagnostic test used, the true prevalence in this region is 93.71% (95%CI: 89.98–97.44%). When analysing flocks, it was verified that 92 ovine flocks (85.98%), 32 caprine flocks (81.25%), and 11 mixed flocks (100%) were positive for SRLVs.

In each flock, an average of 17 (17.38 ± 1.28) blood samples were collected. The distribution of flocks was as follows: 21 flocks (14.00%) did not have any positive animal; 7 (4.67%) had less than 10% of positive animals; 66 (44.00%) had between 10 and 50%; 49 (32.67%) had between 50 and 90%; and 7 (4.67%) more than 90% of positive animals. From a total of 2607 collected samples, 1047 showed positive results in the diagnostic test; therefore, the estimated prevalence was 41.20% (95%CI: 39.32–43.07%), and the actual prevalence was 44.26% (CI 95%: 42.36–46.15%). By species, 778 ovines (38.23%) and 296 caprines (51.75%) were positive.

### 3.2. Risk Factors Analysis

Numerous factors that could influence SRLV infection in small ruminants in this region of Portugal were analysed and are shown in Table 2. These potential risk factors were identified using a questionnaire for small ruminant producers in the region.

Univariate risk factor analysis found a statistically significant association between seropositivity to SRLVs and species (caprine: *p* < 0.0001; OR = 1.73, 95%CI: 1.44–2.09), age (>2 years old: *p* < 0.0001; OR = 2.15, 95%CI: 1.80–2.55), flock size (>100 animals: *p* < 0.0001; OR = 1.60, 95%CI: 1.36–1.86), production system (intensive: *p* < 0.0001; OR = 5.29, 95%CI: 2.77–10.07), food production system (milk: *p* < 0.0001; OR = 1.73, 95%CI: 1.47–2.04), type of activity (professional: *p* < 0.0001; OR = 2.21, 95%CI: 1.71–2.84), participation in livestock competitions (yes: *p* = 0.018; OR = 1.33, 95%CI: 1.05–1.68), replacement young ewe bought (yes: *p* < 0.0001; OR = 1.60, 95%CI: 1.31–1.94), and natural feeding management (yes: *p* = 0.0375; OR = 1.89, 95%CI: 1.03–3.44).

No statistically significant association (*p* > 0.05) was found between seropositivity to SRLVs and breed, mixed flocks, contact with other flocks, mating with males from other flocks, and unhealthy animal isolation. Other factors, despite presenting a statistically significant association, may act as confounding factors: a producer with training in animal production (yes: *p* = 0.0372), a producer who knows the disease (yes: *p* < 0.0001), artificial insemination performed (yes: *p* < 0.0001), and regular veterinary assistance (yes: *p* < 0.0001). Regarding these risk factors, the available bibliography does not provide support or an accurate explanation, and they should be studied in future studies [16,49].

## 4. Discussion

Limited data about SRLV prevalence in sheep and goat populations in Portugal have been published. This study demonstrates that SRLV infection is widespread in the north region of Portugal, affecting 86% of the participating flocks and about 41% of the sampled animals. In sheep flocks, the verified seroprevalence was 85.98% of positive flocks and 38.23% of positive animals. A study carried out in 1995 in Portugal showed a slightly higher prevalence for the region. In this study, a smaller sample size and other diagnostic laboratory tests were used, making it difficult to compare with [50]. There is some variation in prevalence data presented in the literature from different regions in comparison with those obtained in our research. In some studies in Spain, a neighbouring country, a similar prevalence in sheep has been reported [5,46,51]. However, other studies have also reported a lower prevalence [3,8,52]. In other continents, prevalence tends to be lower than in Europe [53,54].

In goats, the prevalence obtained in our study was 81.25% of positive flocks, and the individual prevalence was 51.75%. Some studies reported a similar prevalence in goat flocks [49], and others reported a lower one, specifically for individual prevalence [1,6,55,56,57]. Some of the lower levels of seroprevalences reported in some studies are due to official or voluntary control programs implemented in these countries. It is important to mention that Portugal never had an official program to control this disease. Some more developed farms in other regions of the country started individual programs on their own initiative with the help of their veterinarians.

Sheep and goat rearing in the north region of Portugal are mainly semi-extensive, with grazing during the day and collection at night in stables or high-density fences. More traditional production methods prevail, and management practices are very standardised. The statistical significance analysis carried out in this study demonstrates that certain risk factors can influence the presence of SRLV infection in flocks of small ruminants in this region.

Species analysis showed to have an association with SRLV infection. Goat flocks showed higher seroprevalence and a greater probability of occurrence of the infection than sheep. Phylogenetic studies are necessary to know the SRLV variants circulating in the region. Some of these appear to be species-specific; however, others transmit between both species [58,59]. Some studies point to breed as a possible risk factor for SRLV infection [49,51,60]. There is evidence that host genetics (breed) may influence its susceptibility/resistance to SRLV infection and disease progression [61,62]. In our region, there are four indigenous sheep breeds and two indigenous goat breeds; for this reason, we only check whether the analysed flock had one of the indigenous breeds or an exotic breed. We found no statistical association between this distinction and infection. Regarding animals’ age, we found that those over 2 years old had a significantly higher seroprevalence and were more than twice as likely to be infected. This is in concordance with many studies that reported age as a relevant risk factor [3,54,60]. This may be due to lifetime exposure to the agent that can determine the contagion of animals free of infection at some point [63]. It is added that late seroconversion, characteristic of this disease, can also influence laboratory positivity and delay diagnosis [9,14].

Flock size also indicates to be statistically associated with SRLV infection. Flocks with more than 100 animals were more likely to acquire the infection than those with less than 100. These data have been reported in several other epidemiological studies [5,8,47,54,55]. Similarly, intensively reared animals also had a significantly higher seroprevalence and a greater probability of infection. Both risk factors are mentioned to have a relevant influence on SRLV infection in the literature [51,52,60]. It is common that larger flocks are also produced more intensively, with greater population density, facilitating the transmission of the virus between animals [13,46]. We also obtained higher seroprevalences in dairy flocks compared to meat production flocks. The literature needs to be more precise about the influence of productive aptitude. However, it is known that the productive pressure on dairy sheep is much higher than on meat-production flocks. Lactating ewes may be immunologically compromised and susceptible to various infections, including SRLVs [13]. The literature indicates that SRLVs can infect mixed sheep and goat flocks more frequently than single-species flocks [3,5,54]. However, in our study, there was no statistical association between mixed flocks and their positivity. As previously mentioned, phylogenetic studies are needed in this region for a better understanding of SRLV variants present that may influence these data.

Regarding producers, we found that those with training in livestock production and those who knew the disease had higher seroprevalence, which seems like contradictory data. However, we can speculate that trained producers tend to have larger flocks and an intensive regimen. Additionally, producers who knew about the disease could have been affected by it in their flock and probably were previously diagnosed by their veterinarian. The percentage of producers who do not know about the disease is also high in other studies [8], and, therefore, not knowing about the disease and not being motivated to fight it. It was also found that professional producers had a higher prevalence and more than twice the probability that their animals were infected compared to hobby producers. Despite greater knowledge and attention on the part of professional producers, they usually have larger flocks and often trade in animals, which can contribute to higher prevalence. Participation in livestock competitions was also shown to be statistically associated with SRLV infection. Livestock contests favour the permanence, in the same space, of animals with different origins and unknown health statuses for some diseases. It is added that these also favour the trade of breeders between flocks, which will enter the farm without worrying about screening for infection [64].

Contact between different small ruminant flocks was not associated with infection in this study. Despite this, we know that it can play an important role in the dissemination of several diseases, including SRLVs [54]. This is a concern in this region, where flocks are often driven through common pastures and spend the night in urban areas where other flocks may also be held, posing a risk [47]. Buying animals from other flocks may also pose a risk of diseases entering the flock [57]. The purchase of replacement young ewes showed a higher seroprevalence and a higher probability of infection in our study. Similarly, natural feeding management was also significantly associated with SRLV infection. Lambs and kids that suck colostrum and milk from positive females are one of the most effective means of transmission considered in the bibliography [65]. We found that performing artificial insemination was significantly associated with the disease. These data are not in line with the literature that mentions natural breeding as a possible risk factor [15,49]. Although there is no clear evidence of venereal transmission [17], artificial insemination is usually performed using SRLV-free semen whenever purchased from certified centres. This factor presented in our results may only act as a confounding factor, or the artificial insemination practised does not follow the most appropriate norms. Farms that had regular veterinary care had a higher prevalence of infection. This contradictory fact may not be accurate because these flocks are usually also larger and have more relevant productive pressures.

Other risk factors commonly presented in the literature are difficult to analyse in this region. The standardisation of flocks’ characteristics and management carried out makes it difficult, on one hand, to collect other types of data and, on the other hand, reduces the robustness of the potential risk factors that we present.

This study has potential limitations. First, the research design is one, since it is a cross-sectional study that was carried out at a local level in a single region. It is our future goal to include other regions of Portugal to obtain information about SRLV infection in these other regions as well as in different production systems. This study may also present a type −1 error due to the high number of variables included in the model and the number of statistical tests performed. Performing a multivariate analysis instead of a univariate one could also provide greater robustness to our data. Due to these limitations imposed by the study design itself, these results need to be interpreted with care, as it was not possible to clearly identify a cause-and-effect relation.

High seroprevalence verified in this study supports the urge to develop a strategy for implementing effective SRLV control programs. Due to the high costs of implementing an exhaustive control program, initially, the reduction and minimisation of the risk of infection by SRLVs should be promoted through biosecurity measures such as (i) removal of offspring from mothers soon after birth and artificial rearing; (ii) separation of infected animals; (iii) periodic screening for SRLVs; and (iv) acquisition of animals from certified SRLV-free flocks. Later, more drastic measures, such as the culling of seropositive animals, can be implemented, but they are only viable for a low prevalence of infection. However, the motivation of producers is essential for the success of a possible control program. The immediate economic and productive benefits of controlling this disease should be highlighted.

## 5. Conclusions

In this study, we found a high SRLV seroprevalence in sheep and goat flocks, concluding that SRLV infection is widespread in small ruminant flocks throughout the northern region of Portugal. The epidemiological study of risk factors contributes to a greater and better knowledge of the disease. Early detection of this disease is essential, using laboratory tests such as serological tests. Thus, adapted and effective preventive measures can be implemented to reduce viral transmission. This study is a model to encourage veterinary health authorities to promote and audit voluntary control and eradication programs to control this disease in sheep and goat flocks in Portugal.

## Figures and Tables

**Table 1 pathogens-12-00829-t001:** SRLV Seroprevalence of individuals and flocks in the north region of Portugal.

	Flocks		Animals	
	Analysed (*n*)	Positive (%)	Analysed (*n*)	Positive (%)
Sheep	107	92 (85.98)	2035	778 (38.23)
Goats	32	26 (81.25)	572	296 (51.75)
Mixed	11	11 (100)	-	-
Total	150	129 (86)	2607	1074 (41.20)

**Table 2 pathogens-12-00829-t002:** Potential risk factors associated with SRLV infection in the North of Portugal.

Variable	Analysed (*n*)	Seroprevalence (%)	*p* Value	Odds Ratio
**Species**			<0.0001	1.73 (1.44–2.09)
Caprine	572	296 (51.75)		
Ovine	2035	778 (38.23)		
**Breed**			0.6898	-
Exotic	1415	588 (41.55)		
Autochthonous	1192	486 (40.77)		
**Age**			<0.0001	2.15 (1.80–2.55)
>2 years old	1735	818 (47.15)		
<2 years old	872	256 (29.36)		
**Flock size**			<0.0001	1.60 (1.36–1.86)
>100 animals	1572	718 (45.67)		
<100 animals	1035	356 (34.40)		
**Production system**			<0.0001	5.29 (2.77–10.07)
Intensive	55	43 (78.18)		
Semiextensive	2552	1031 (40.40)		
**Food production system**			<0.0001	1.73 (1.47–2.04)
Milk	868	435 (50.12)		
Meat	1739	639 (36.75)		
**Mixed flock**			0.2239	-
Yes	192	71 (36.98)		
No	2415	1003 (41.53)		
**Producer with training in animal production**			0.0372	1.25 (1.01–1.54)
No	2174	876 (40.29)		
Yes	433	198 (45.73)		
**Producer knows the disease**			<0.0001	2.13 (1.71–2.65)
Yes	382	218 (57.07)		
No	2225	856 (38.47)		
**Type of activity**			<0.0001	2.21 (1.71–2.84)
Professional	2256	983 (43.57)		
Hobby	351	91 (25.93)		
**Participation in livestock competitions**			0.0180	1.33 (1.05–1.68)
Yes	319	151 (47.34)		
No	2288	923 (40.34)		
**Contact with other flocks**			0.0564	-
Yes	1561	667 (42.73)		
No	1046	407 (38.91)		
**Replacement young ewe bought**			<0.0001	1.60 (1.31–1.94)
Yes	495	250 (50.51)		
No	2112	824 (39.02)		
**Natural feeding management**			0.0375	1.89 (1.03–3.44)
Yes	2552	1059 (41.50)		
No	55	15 (27.27)		
**Artificial insemination performed**			<0.0001	7.75 (4.05–14.86)
Yes	68	57 (83.82)		
No	2539	1017 (40.06)		
**Mating with males from other flocks**			0.1784	-
Yes	200	73 (36.50)		
No	2407	1001 (41.59)		
**Unhealthy animals’ isolation**			0.8627	-
No	1815	750 (41.32)		
Yes	792	324 (40.91)		
**Regular veterinary assistance**			<0.0001	2.39 (1.84–3.10)
Yes	263	159 (60.46)		
No	2344	915 (39.04)		

## Data Availability

Data sharing not applicable.

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
