# Peer review of "Small Ruminant Lentivirus Infection in Sheep and Goats in North Portugal: Seroprevalence and Risk Factors"

_pathogens, 2023, doi:10.3390/pathogens12060829_

Round 1

Reviewer 1 Report

In the manuscript entitled “Small Ruminant Lentivirus infection in sheep and goat in North Portugal: Seroprevalence and risk factors”, the author determined the seroprevalence of lentivirus along with its risk factors. However, some problems should be revised carefully.

Ø  The author has mentioned the virus as Small Ruminant Lentivirus (SRLV) throughout the manuscript. It would be appropriate to mention as small ruminant lentiviruses as small ruminant lentiviruses include caprine arthritis encephalitis virus (CAEV) and visna/maedi virus (VMV), also called ovine progressive pneumonia virus (OPPV).

Ø  The sensitivity and specificity of the ELISA used should be mentioned in materials and methods section.

Ø  In materials and methods section, study sites should be discussed in detail. I suggest the authors to depict the same with the help of map including GPS coordinates.

Ø  Line 178: Mention as “true prevalence” instead of actual prevalence.

Ø  What was the basis for choosing the risk factors. Was it based on any previous study. Please cite a reference.

Ø  In age, why the authors have categorized the animals as more than 2 and less than 2. As the market age of small ruminants is less than a year, it should have categorized in a different way.

Ø  What do you mean by production aptitude? Reword.

Ø  What do you mean by artificial rearing and Natural rearing. Does it not sound non-scientific?

Ø  Line 208-209: “Other factors, despite presenting statistically significant association, may act as confounding factors” Explain?

Ø  I suggest the authors to compute multivariate analysis for the risk factors which were statistically associated with seropositivity.

Ø  Table 2: For certain risk factors, the odds ratio was not furnished. Why?

Ø  Line 20 and Table 2: Species

Ø  Line 182: blood samples were collected. “Not drained”

Ø  Line 187: It is 95% CI

Ø  Throughout the manuscript, replace specie with species

Ø  Line 17: Don’t start the sentence with “collected”

Extensive editing of English language required

Author Response

  1. The author has mentioned the virus as Small Ruminant Lentivirus (SRLV) throughout the manuscript. It would be appropriate to mention as small ruminant lentiviruses as small ruminant lentiviruses include caprine arthritis encephalitis virus (CAEV) and visna/maedi virus (VMV), also called ovine progressive pneumonia virus (OPPV).

A – Thank you very much for your constructive comments. The suggestion was included in the manuscript.

  1. The sensitivity and specificity of the ELISA used should be mentioned in materials and methods section.

A – Thank you very much for your constructive comments. The information was included in the manuscript.

  1. In materials and methods section, study sites should be discussed in detail. I suggest the authors to depict the same with the help of map including GPS coordinates.

A – Thank you very much for your constructive comments. We would like to include a map as mentioned but we are limited by the European and Portuguese protection of personal data law. Not all farmers have given us authorization to disclose the coordinates on their farm, so we chose not to include a map in the manuscript. Thanks for your suggestion and understanding.

  1. Line 178: Mention as “true prevalence” instead of actual prevalence.

A – Thank you very much for your constructive comments. The correction has been made throughout the manuscript.

  1. What was the basis for choosing the risk factors. Was it based on any previous study. Please cite a reference.

A – Thank you very much for your constructive comments. This data was included in manuscript. After reviewing the literature and identifying possible risk factors for SRLV infection, a questionnaire was made (please see references 1, 49, 63). The questionnaire included 40 questions, that were close-ended with choices available.

  1. In age, why the authors have categorized the animals as more than 2 and less than 2. As the market age of small ruminants is less than a year, it should have categorized in a different way.

A – Thank you very much for your constructive comments. We carried out a statistical analysis with different ages, although we only verified statistically significant differences in animals older than 2 years. A late seroconversion could explain this situation as mentioned in the manuscript.

  1. What do you mean by production aptitude? Reword.

A – Thank you very much for your constructive comments. The concept has been reworded for “food production system” (i.e., milk or meat).

  1. What do you mean by artificial rearing and Natural rearing. Does it not sound non-scientific?

A – Thank you very much for your constructive comments. The concept has been reworded for “Natural feeding management”. With the options Yes and No, according to natural or artificial feeding management before weaning of lambs/kids, respectively.

  1. Line 208-209: “Other factors, despite presenting statistically significant association, may act as confounding factors” Explain?

A – Thank you very much for your constructive comments. Some risk factors with statistically significant association do not have a plausible explanation.

  1. I suggest the authors to compute multivariate analysis for the risk factors which were statistically associated with seropositivity.

A – Thank you very much for your constructive comments. This is one of the limitations of our study. Our team continues to develop studies in other regions of the country and intends to include this analysis in future comparative studies between regions and production systems.

  1. Table 2: For certain risk factors, the odds ratio was not furnished. Why?

A – Thank you very much for your constructive comments. The OR of the missing risk factors was added. Regarding these risk factors, the available bibliography does not provide support or an accurately explanation, and they should be studied in future studies.

  1. Line 20 and Table 2: Species

A – Thank you very much for your constructive comments. The correction has been made.

  1. Line 182: blood samples were collected. “Not drained”

A – Thank you very much for your constructive comments. The correction has been made.

  1. Line 187: It is 95% CI

A – Thank you very much for your constructive comments. The correction has been made.

  1. Throughout the manuscript, replace specie with species.

A – Thank you very much for your constructive comments. The correction has been made.

  1. Line 17: Don’t start the sentence with “collected”.

A – Thank you very much for your constructive comments. The correction has been made.

Reviewer 2 Report

The manuscript entitled "Small Ruminant Lentivirus Infection in Sheep and Goats in North Portugal: Seroprevalence and risk factors" presents results of a seroprevalence and associated risk factors for SRLV infection in the Bragança district, a restricted region of northern Portugal. 

The type of sampling, carried out in a single and specific area of Portugal, although numerically adequate, could unfortunately be influenced by specific conditions related to the study area and/or the probable presence and evolution of the pathogen in the area and the local small ruminant population, making the risk analyses carried out less robust.

Furthermore, a phylogenetic study of the viruses present in the area is completely lacking, which could make missing many aspects related to distribution and epidemiology. In conclusion, even if the innovative contribution and the contribution of the research are modest, the research may still be of interest to open future studies on the presence and characteristics of SRLV infections in the territory.

some minor details:

Line 14: Delete “are infected”

Line 17: Samples were collected from a total of 150 flocks, of which 129 (86.0%; 95% CI: 80.67% - 91.33%) had at least one sero-positive animal.

Line 36: SRLVs

Line 40: The seroprevalences reported in different studies are difficult to compare because of differences in the sensitivity and specificity of the diagnostic tests used as well as the criteria used to…

Result: remove from the total number of the herds the lost one, consider only  150 herds

Line 253 please provide reference or evidences for this sentence “It is added that the late seroconversion, characteristic of this disease, can also influence the laboratory positivity and delay the diagnosis”

English could be improved.

Author Response

REVIEWER # 2

  1. The type of sampling, carried out in a single and specific area of Portugal, although numerically adequate, could unfortunately be influenced by specific conditions related to the study area and/or the probable presence and evolution of the pathogen in the area and the local small ruminant population, making the risk analyses carried out less robust.

A – Thank you very much for your constructive comments.

  1. Furthermore, a phylogenetic study of the viruses present in the area is completely lacking, which could make missing many aspects related to distribution and epidemiology. In conclusion, even if the innovative contribution and the contribution of the research are modest, the research may still be of interest to open future studies on the presence and characteristics of SRLV infections in the territory.

A – Thank you very much for your constructive comments. Our team continues to develop studies in other regions of the country and intends to include these suggestions in future manuscripts.

  1. Line 14: Delete “are infected”.

A – Thank you very much for your constructive comments. The correction has been made.

  1. Line 17: Samples were collected from a total of 150 flocks, of which 129 (86.0%; 95% CI: 80.67% - 91.33%) had at least one seropositive animal.

A – Thank you very much for your constructive comments. The correction has been made.

  1. Line 36: SRLVs

A – Thank you very much for your constructive comments. The correction has been made.

  1. Line 40: The seroprevalences reported in different studies are difficult to compare because of differences in the sensitivity and specificity of the diagnostic tests used as well as the criteria used to…

A – Thank you very much for your constructive comments. The correction has been made.

  1. Result: remove from the total number of the herds the lost one, consider only 150 herds.

A – Thank you very much for your constructive comments. The lost flock has been removed.

  1. Line 253 please provide reference or evidences for this sentence “It is added that the late seroconversion, characteristic of this disease, can also influence the laboratory positivity and delay the diagnosis”

A – Thank you very much for your constructive comments. We apologize for the mistake; the reference has been added.